# Assessment of Recreational and Cultural Ecosystem Services Value of Islands

**Na Zhao [1,2], Hui Wang [1,*], Jingqiu Zhong [1,3,4] and Dongqi Sun [4]**

1   School of Geography, Liaoning Normal University, Dalian 116029, China; zhaona2630@hlbrzy.com (N.Z.); zjq@lnnu.edu.cn (J.Z.)
2   Department of Business and Tourism, Hulunbuir Vocational Technical College, Hulunbuir 021000, China
3   Center for Studies of Marine Economy and Sustainable Development, Liaoning Normal University, Dalian 116029, China
4   State Key Laboratory of Resources and Environmental Information System, Institute of Geographic Sciences and Natural Resources Research, CAS, Beijing 100101, China; sundq@igsnrr.ac.cn
*   Correspondence: wanghui@lnnu.edu.cn; Tel.: +86-13840841336

**Abstract:** With the gradual expansion of benefits provided by islands to human wellbeing, it has become increasingly important to quantify the cultural ecosystem service functions of islands. In this study, the zone travel cost method (ZTCM) and individual travel cost method (ITCM) are used to assess the recreational and cultural ecosystem services value of the island region of Changhai County, China, and the recreational value of Xiaochangshan, Dachangshan, and Guanglu Islands based on questionnaire survey data. The results are as follows: (1) The overall recreational and cultural ecosystem services value of Changhai County in 2019 was 533.14 million USD, including a traveling cost of 395.71 million USD and consumer surplus of 137.43 million USD. (2) Further, based on the choice and consumption behaviors of tourists, multiple recreational values of the county were also inferred: the aesthetic experience presented the greatest contribution, while educational knowledge accounted for a lower percentage. (3) The recreational and cultural ecosystem services value 294.16, 60.03, and 143.26 million USD for Xiaochangshan, Dachangshan, and Guanglu Islands, respectively. Based on the findings of this study, future research will focus on the planning and development of Xiaochangshan Island tourism to maximize consumer surplus.

**Keywords:** travel cost method (TCM); cultural ecosystem services (CES); recreational value; Changhai County

## 1. Introduction

An island is a naturally formed land mass surrounded by water that exists above water at high tide [1]. Islands play an important strategic role in safeguarding national rights and interests, defense and ecological security, and economic and social development [2]. As an important part of the marine ecosystem, islands are rich in tourism resources and have become attractive tourist destinations with their unique charm [3]; this island tourism has attracted the attention of scholars [4–7]. However, tourism also has a negative impact on the island ecosystem service function [8]. With the arrival of many tourists, the provisioning, regulating, and supporting functions of island ecosystem services have increasingly degraded, resulting in a lack of freshwater resources, and increased resource consumption, pollutant discharge, severe flooding, soil erosion, and bare rock area [9–11]. Simultaneously, the ecological environment has suffered severe damage, and the human–island relationship has been adversely affected. Despite this, island culture remains a major tourist attraction and a unique selling point.

Previous studies have focused on the tourism industry, society, environment, and sustainable aspects of island tourism research. Bater (1997) regarded the sustainable development of the island economy as a starting point and proposed that the balanced

development of the economy should be paid attention to under the premise of limited resources and population increase [12]. Abeyratne (1999) believed that the region must fully consider the impact of tourism development on the economy, social culture, environment, and ecology, as well as regional cooperation, marine science, and technological application, in addition to several other issues to ensure the moderate and sustainable development of tourism [13]. Kokkranikal et al. (2003) studied the sustainable development of tourism in Loksawai Islands, India. They pointed out that under the influence of geographical location and environmental constraints, the premise of sustainable development is to turn disadvantages into advantages and promote sustainable development of tourism by restricting people's behaviors [14]. Kerr (2005) explored the development and characteristics of small islands, compared different sustainable development models, and discussed the sustainable development of small islands with empirical research [15]. A study conducted by Méheux et al. (2006) on Vanuatu Island in the western Pacific showed that whether tourism managers are aware of natural hazards is of decisive significance in the business process, and their psychological preparation for this can better safeguard the sustainable development of the island [16]. The above studies have qualitatively analyzed the relationship between island tourism and the ecological environment. Compared to the mainland, islands have more fragile ecological environments and less resilience. In order to better balance the relationship between island environmental protection and tourism development, it is essential to evaluate the value of island recreation from a quantitative perspective. Scientific assessment of recreational and cultural ecosystem services values of islands can provide data to support the optimization of resource allocation and development and utilization efficiency. This study aims to improve the government's understanding of island ecological environments in a more comprehensive manner. Decision-makers and planners should consider the ecological value of the natural environment and be aware of the environment's recreational and cultural values. Our findings suggest that the recreational and cultural ecosystem services value of islands can be significantly enhanced by improving island facilities, deepening the cultural value of recreational activities, and reflecting the practical significance of this research.

Ecosystem services are important links and bridges connecting the natural environment and human wellbeing [17]. Therefore, effective measurement and monitoring ecosystem services are critical for promoting harmony between humans and nature, as well as practicing the concept of ecological civilization construction [18]. Ecosystem services refer to the various benefits of tangible materials and intangible services directly or indirectly obtained from the ecosystem [19]. The Millennium Ecosystem Assessment (in 2005) divided ecosystem services into four categories: supporting, regulating, provisioning, and cultural services. Among these, cultural services are defined as those non-material benefits humans obtain from the ecosystem through spiritual satisfaction, cognitive development, thought, amusement, and aesthetic experience [20]. The concept of cultural services can be further expanded to include tourism recreation, aesthetic experience, spiritual satisfaction, social relations, and educational knowledge [21]. Tourism recreation refers to the leisure, entertainment, and cultural services provided by the ecosystem based on natural landscapes, which have economic, social, and ecological benefits [22,23]. Recreation is characterized by non-exclusive and non-competitive quasi-public goods [24], and its value is essentially a manifestation of the economization of tourism resources [25]. The assessment of the recreation economic value proposed by Costanza is an important accounting element in ecosystem services [26], and serves as the theoretical basis for fixing the reasonable consumption price in tourist attractions, along with the effective development and protection of tourism resources [27]. Since the 1950s, the evaluation of recreational value has been an important research direction in economics, geography, and environmental science [28,29]. In the 1960s and 1970s, with the rapid development of the global tourism industry and increasing awareness of environmental conflict, environmental economists began to notice that natural resources and the environment were not limitless, and, realizing the scarcity of resources, to consider the economic value of non-market goods or services; consequently,

research turned to the evaluation of monetary value [30]. Since the 1980s and 1990s, the field has witnessed the formation of various methods of determining value: the public goods theory; the theory of western and welfare economics; consumer surplus, opportunity cost, willingness to pay, resources and environment value theory; the continuous improvement of the public goods theory; and the recreation economic value evaluation theoretical system [31]. Therefore, the research methods of determining recreational value are becoming increasingly diversified.

As a non-market item, recreational value cannot be directly assessed for its economic value; its market value can only be estimated by substituting other valuable items or assessing the behavior of consuming related commodities [32]. The travel cost method (TCM) and contingent valuation method (CVM) are the main research methods used for recreational value assessment. Both models are based on the utility value and consumer surplus value theories. The main difference between them is that the TCM uses the actual consumption of tourists to calculate consumer surplus, whereas the CVM is a stated preference method that calculates consumer surplus according to the consumption willingness of tourists [33–36]. CVM is greatly disturbed by tourists' subjective factors [37]. Based on the research objects and contents, TCM has been more widely used and improved by domestic and foreign research scholars [38,39]. In this study, TCM is used to analyze the recreational and cultural ecosystem services value of islands. In 1947, Hotelling [40] first proposed the TCM, and then Clawson [41] conducted more in-depth research on the travel cost model. Mendelsohn et al. studied the steps and methods of TCM [42]. Finally, in the 1980s, TCM was introduced to the domestic research field [43]. The method is mainly used in national parks, forests, wetlands, and lakes containing recreational functions that feature scenic spots, nature reserves, and cultural tourism attractions [44]. There are few studies on island recreation value using TCM. China is one of the countries with the largest number of islands. The island distribution is wide, and the island types are diverse. This study uses TCM to evaluate the recreational value of islands in China, which is one of the innovations.

There are two common types of TCM models: zone TCM (ZTCM) and individual TCM (ITCM). The former assumes that the consumption behaviors of all people in a tourist source area are the same, whereas the latter assigns more weightage to individual consumption behavior [45–47]. Clawson and Knetsch proposed the ZTCM [48], wherein tourists in the recreational area are divided into different tourist source areas, and their travel rate and total travel cost for each tourist source area are calculated. Subsequently, once the consumption demand function of each tourist source area is obtained, the consumer surplus of each tourist source area is calculated and summed, and the total value of consumer surplus of the recreation area is obtained [49]. ITCM was proposed by Brown and Nawas [50] and does not need to divide travel communities. Rather, it considers the number of visits of each tourist as a function of travel time and other explanatory variables [51]. Due to the different research emphases of ZTCM and ITCM, this study used ZTCM to analyze the overall recreational and cultural ecosystem services value of Changhai County, and ITCM to compare and analyze the recreational and cultural ecosystem services value of Xiaochangshan, Dachangshan, and Guanglu Islands. Another innovation of this study is to analyze the recreational and cultural ecosystem services value of islands from different perspectives using two methods.

The theoretical significance of this study lies in that Changhai County is considered as the starting point to study the evaluation method of island ecosystem recreation value, which enriches theoretical research in this field. Additionally, it allows the ecosystem's recreational value to be determined more accurately and provides a scientific basis for island tourism ecological compensation.

This study expands the scopes of ZTCM and ITCM and applies them to islands. The remainder of the study is divided into three parts; Section 2 describes the island county study region and three main islands of focus and outlines the TCM theories applied in the present study. Section 3 presents the results. Section 4 details policy recommendations based on the results of the study and lists the research limitations and prospects.

## 2. Materials and Methods

### 2.1. Study Area

Changhai County (38°55′–39°35′ N, 122°17′–123°13′ E) is located in the northern Yellow Sea on the east side of the Liaodong Peninsula, China. It belongs to Dalian City of Liaoning Province and is adjacent to the Jinzhou District, Pulandian County-level City, and Zhuanghe County-level City, which are components of the larger Dalian City region. Changhai County is the only island county in Northeast China and the only island border county in China. With favorable climatic conditions, a beautiful ecological environment, and abundant resources, Changhai County provides several services, such as food supply and climate regulation, and cultural services, such as sightseeing and supporting human traditional culture emotions. In 2010, the Dalian Municipal Government approved and established the Changshan Islands Marine Ecological Economic Zone, and in 2014, the State Oceanic Administration (also known as the Ministry of Natural Resources) established the Dalian Changshan Islands National Marine Park. More recently, in 2016, the Changhai County Party Committee launched the construction of an international ecological island. Changhai County has been generally developed as an eco-tourism island, integrating leisure meeting areas, environmental improvement projects, fishing culture experiences, recreational sports, amusement parks, nature tours, and holiday residence. In 2019, Changhai County received a total of 1.34 million tourists on the island and achieved a comprehensive tourism income of 241.02 million USD, with an annual increase of 0.70% and 7%, respectively [52].

The county comprises 252 islands, with a total land area of 142 km$^2$, a total sea area of 10,324 km$^2$, and a total coastline length of 359 km. It governs five towns, namely Dachangshan, Xiaochangshan, Guanglu, Zhangzi, and Haiyang Islands. Guanglu, Dachangshan, and Xiaochangshan are its key development islands. The Changhai County Government is on Dachangshan Island, which is centrally located in the Changshan Archipelago. The island has many natural and cultural landscapes, with the main tourist attractions being Qixiang Garden, Sanyuan Palace, Beihai Bathing Beach, and the Memorial Tower of Island Maker and Protector. Xiaochangshan Island is connected to Dachangshan Island via the 1790 m Changshan Bridge and enjoys its reputation as a "Natural Fishery Village", because the island has excellent fishing conditions and has been approved by the China Angling Association as a national sea fishing base. Consequently, Xiaochangshan Island attracts many Chinese and foreign fishing enthusiasts and tourists every year. Guanglu Island is located west of the Changshan Archipelago and is the archipelago's largest island. The main tourist attractions of Guanglu Island are Mazu Temple, Xiaozhushan Ruins, and Bathing Beach. Therefore, this study selected Dachangshan, Xiaochangshan, and Guanglu Islands as the main study areas (Figure 1).

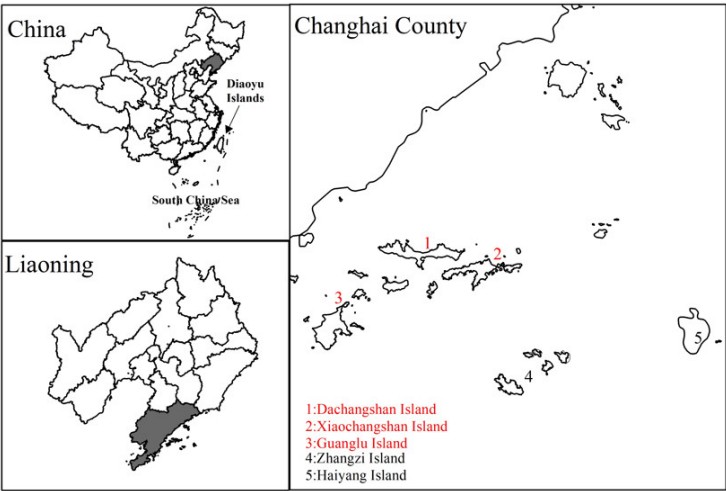

**Figure 1.** The location of the study area (Changhai County, China).

### 2.2. Methodology

2.2.1. Questionnaire Method

Primary data were collected using a questionnaire survey divided into two parts. The first part only requested basic information on the tourists, including gender, age, education, monthly income, and source area. The second part included the recreational characteristics of tourists, including traveling cost, transportation, number of trips, length of stay, partner tours, and purpose of visit. Prior to the main survey, the research group conducted online and offline pre-investigations to identify any insufficiencies or flaws in the questionnaire design and facilitate the continuous adjustment of the questionnaire contents and interview methods and improve the formal questionnaire content. In July and August 2019, the research group conducted face-to-face interviews with tourists on Dachangshan, Xiaochangshan, and Guanglu Islands. The survey method adopted non-probability sampling and the random encounter method. To ensure the representativeness of the samples, the group consciously avoided selecting tourists from the same tour group or fellow travelers. Each questionnaire was filled out and collected on the spot [53].

2.2.2. Zone Travel Cost Method (ZTCM)

The ZTCM is the most traditional travel cost model. This study used the ZTCM to assess the recreational and cultural ecosystem services value of Changhai County. The basic steps are described as follows [54,55]:

- Divide the tourist source area and calculate the travel rate

The first step of the ZTCM model is to determine the tourist source region and calculate the travel rate of each tourist source area, which was carried out using Equation (1):

$$R_i = \frac{V_i}{P_i} = \frac{(n_i/N) \times V}{P_i}, i = 1, 2, \ldots, k.$$ (1)

where $R_i$ is the travel rate from the tourist source area $i$ to the study area; $P_i$ is the urban population of the tourist source area $i$; $V_i$ is the number of annual visits from the tourist source area $i$ to the study area; $n_i$ is the actual sample number of the tourist source area $i$; $N$ is the total number of survey samples; $V$ is the annual tourist number received in the study area; and $k$ is the number of tourist source areas.

- Calculate traveling costs in each tourist source area

The traveling costs include both explicit and implicit parts. Explicit costs are the travel cost obtained from the questionnaire; implicit costs refer to the opportunity costs of time, including traffic time and stay time. Based on existing research, the opportunity cost of time per unit was calculated using one-third of the wage rate, assuming 250 working days per year [56]. The total traveling cost of tourists can be expressed as:

$$C_i = W_i + \frac{1}{3} \times \frac{Y_i}{250} \times D_i, \ i = 1, 2, \ldots, k.$$ (2)

where $C_i$ is the per capita travel cost of tourists in the tourist source area $i$; $W_i$ is the actual per capita explicit costs; $Y_i$ is the disposable income of the residents in the tourist source area $i$; $D_i$ is composed of two parts: traffic time and stay time; and $k$ is the number of tourist source areas.

- Draw the tourism demand curve

According to the general demand theory, with travel rate as the dependent variable, per capita travel cost, and other social indicators as independent variables, regression was performed to obtain the first-stage demand function. The formula is as follows:

$$R_i = f(C_i, Y_i, \cdots, D_i)$$ (3)

where $R_i$ is the travel rate from tourist source area $i$ to study area; $C_i$ is the per capita travel cost of tourists in the tourist source area $i$; and $Y_i, \cdots, D_i$ are other social indicators.

- Calculate consumer surplus

Taking the number of tourists after the additional cost as the dependent variable and the additional cost as the independent variable, the second-stage tourism demand fitting model was established and integrated to obtain the consumer surplus in the study area. The formula is as follows:

$$CS = \int_0^{p_m} f(x)dx \tag{4}$$

where $CS$ is the consumer surplus; $p_m$ is the additional cost value when the number of tourists is 0; $f(x)$ is the second-stage tourism demand function model; and $x$ is the added travel cost.

- Assess the total recreational and cultural ecosystem services value of the study area

Finally, the total value of the ecosystems services of the study area was calculated using the following equation:

$$TV = TC + CS \tag{5}$$

where $TV$ is the total recreational and cultural ecosystem services value; $TC$ is the total traveling cost; and $CS$ is the consumer surplus.

### 2.2.3. Individual Travel Cost Method (ITCM)

The demand function constructed by ITCM can better reflect the tourist inter-individual changes. The consumer surplus of the three main islands was assessed by analyzing the individual behavior of tourists and their travel cost, and then obtaining the recreational and cultural ecosystem services value of each island. The evaluation steps are described as follows [57–59].

- Establish the tourism demand function model

The travel cost and the number of trips for the individual were calculated; then, the factors related to the times of the trip as a model variable were determined, and the tourism demand function model between the number of trips, travel cost, and related factors was developed. The formula is as follows:

$$Q_i = f(P_i, Y_i, D_i, \cdots, S_i) \tag{6}$$

where $Q_i$ is the number of trips for the tourist $i$ to the study area; $P_i$ is the travel cost of the tourist $i$; and $Y_i, D_i, \cdots, S_i$ are other related factors.

- Calculate consumer surplus

According to the demand function model, the individual consumer surplus of tourists was calculated. The calculation formula is as follows:

$$C_s = -\frac{q_0^2}{2\beta_1} \tag{7}$$

where $C_s$ is the individual consumer surplus of the tourist, $q_0$ is the number of trips for the tourist, and $\beta_1$ is the estimated coefficient of travel cost in the demand function.

- Calculate the recreational and cultural ecosystem services value of each island

Finally, the total value of the ecosystem services of each island was calculated using the following equation:

$$T_v = T_c + C_s \tag{8}$$

where $T_v$ is the recreational and cultural ecosystem services value on each island; $T_c$ is the per capita travel cost on each island; and $C_s$ is consumer surplus on each island.

2.2.4. Method Flowchart

The methodology is explained clearly by methods and technical route (Figure 2).

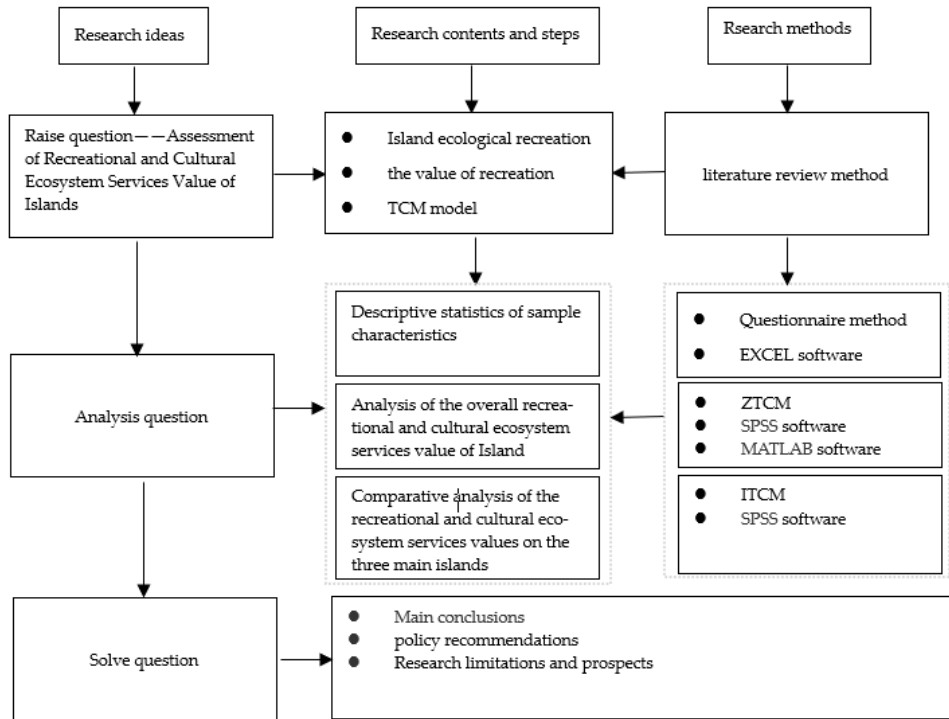

**Figure 2.** Methods and technical route.

**3. Results**

*3.1. Descriptive Statistics of Sample Characteristics*

A total of 315 questionnaires were returned, of which 303 were deemed valid, with a questionnaire effectivity rate of 96.20%. The selected population and society economic characteristics used for this analysis are reported as follows (Table 1). The proportion of female tourists (60.40%) was much higher than that of male tourists (39.60%). In terms of age structure, middle-aged tourists were the main tourist source, accounting for 63.37%. Most of the tourists participating in the survey were either undergraduates, college graduates, or middle-income groups. The number of tourists visiting the island for the first time was relatively high, and 31.35% of tourists had chosen to revisit. In terms of the tourist source area, the surveyed tourists were mainly from Liaoning Province: Dalian, Shenyang, and Anshan, and the difference between total tourists from Liaoning Province and total tourists from outside the province were relatively small.

**Table 1.** Sample description statistical table.

| Variable | Category | Proportion (%) | Variable | Category | Proportion (%) |
|---|---|---|---|---|---|
| Gender | Male | 39.60 | Monthly Income | 300 U or less | 17.49 |
| | Female | 60.40 | | 300–750 | 40.27 |
| Age | <30 | 22.44 | | 750–1200 | 27.72 |
| | 30–50 | 63.37 | | 1200 or more | 14.52 |
| | >50 | 14.19 | Number of trips | 1 | 68.65 |
| Education | Primary and below | 3.30 | | 2–3 | 22.11 |
| | Middle school | 12.87 | | 4–5 | 5.28 |
| | High and secondary | 25.41 | | >5 | 3.96 |
| | Undergraduate/College | 47.86 | Tourist source area | Inside the province | 45.87 |
| | Postgraduate and above | 10.56 | | Outside the province | 54.13 |

### 3.2. Analysis of the Total Recreational and Cultural Ecosystem Services Value of Changhai County

In this research, the province-level administrative regions and prefecture-level cities in Liaoning Province were considered the tourist source region and were divided into 27 tourist source areas. Based on the actual sample number of each tourist source area, the total number of survey samples, and the number of urban populations, the travel rate of each tourist source area was calculated using Equation (1). Then, based on the per capita explicit travel costs, traffic time, and stay time in the survey sample, the travel cost of each tourist source area was calculated according to Equation (2). The total travel cost of tourists in Changhai County in 2019 was approximately 395.71 million USD (Table 2).

**Table 2.** Tourist travel rate and travel cost of each tourist source area of Changhai County in 2019.

| Source Area | Visitor (Per 10 Thousand People) | Urban Population (Per 10 Thousand People) | Travel Rate (‰) | Explicit Cost (USD) | Per Capita Opportunity Cost of Time (USD) | Per Capita Travel Cost (USD) | Total Travel Cost for Each Source Area (Per 10 Thousand USD) |
|---|---|---|---|---|---|---|---|
| Dalian City | 17.69 | 550.90 | 32.11 | 222.68 | 29.56 | 252.24 | 4461.97 |
| Shenyang City | 19.02 | 673.60 | 28.23 | 281.62 | 29.63 | 311.25 | 5918.97 |
| Anshan City | 9.73 | 259.65 | 37.47 | 197.33 | 26.16 | 223.50 | 2174.47 |
| Fushun City | 4.42 | 156.00 | 28.35 | 187.13 | 23.98 | 211.11 | 933.60 |
| Jinzhou City | 3.98 | 165.94 | 23.99 | 237.03 | 18.74 | 255.77 | 1018.01 |
| Panjin City | 1.33 | 105.30 | 12.60 | 311.88 | 23.42 | 335.30 | 444.85 |
| Liaoyang City | 1.77 | 114.30 | 15.48 | 187.13 | 20.90 | 208.03 | 368.00 |
| Tieling City | 1.33 | 128.50 | 10.32 | 261.98 | 18.31 | 280.28 | 371.86 |
| Yingkou City | 0.44 | 157.60 | 2.81 | 261.98 | 22.18 | 284.15 | 125.66 |
| Chaoyang City | 0.44 | 134.30 | 3.29 | 261.98 | 15.25 | 277.22 | 122.60 |
| Huludao City | 0.44 | 125.40 | 3.53 | 306.55 | 23.85 | 330.41 | 146.12 |
| Fuxin City | 0.44 | 103.80 | 4.26 | 240.91 | 16.53 | 257.44 | 113.85 |
| Benxi City | 0.44 | 131.00 | 3.38 | 154.51 | 13.16 | 167.67 | 74.15 |
| Heilongjiang Province | 16.81 | 2268.00 | 7.41 | 309.25 | 27.15 | 336.40 | 5653.33 |
| Jilin Province | 38.48 | 1556.00 | 24.73 | 287.04 | 24.75 | 311.79 | 11,996.34 |
| Jiangsu Province | 0.88 | 5604.00 | 0.16 | 468.79 | 42.40 | 511.18 | 452.13 |
| Beijing | 1.77 | 1863.00 | 0.95 | 287.87 | 47.50 | 335.37 | 593.26 |
| Shanxi Province | 1.77 | 2172.00 | 0.81 | 221.86 | 21.68 | 243.54 | 430.81 |
| Hebei Province | 1.77 | 4264.00 | 0.41 | 368.08 | 24.68 | 392.76 | 694.78 |
| Inner Mongolia | 5.31 | 1589.00 | 3.34 | 233.72 | 23.55 | 257.27 | 1365.31 |
| Henan Province | 0.88 | 4967.00 | 0.18 | 301.12 | 19.09 | 320.21 | 283.22 |
| Yunnan Province | 1.33 | 2309.00 | 0.57 | 371.16 | 24.53 | 395.69 | 524.97 |
| Shandong Province | 1.77 | 6147.00 | 0.29 | 336.83 | 43.42 | 380.24 | 672.64 |
| Anhui Province | 0.44 | 3459.00 | 0.13 | 486.53 | 27.46 | 513.99 | 227.31 |
| Tianjin | 0.44 | 1297.00 | 0.34 | 187.13 | 25.74 | 212.86 | 94.14 |
| Sichuang Province | 0.44 | 4362.00 | 0.10 | 343.11 | 19.89 | 363.00 | 160.54 |
| Ningxia | 0.44 | 405.00 | 1.09 | 303.44 | 31.83 | 335.27 | 148.27 |
| In total | | | | 39,571.18 | | | |

The results show a significant negative correlation between the capita travel cost and travel rate ($p < 0.05$) according to Equation (3). None of the other independent variables could pass the significance test, which is consistent with the current domestic research that generally considers per capita travel cost as the only independent variable [60]. The model summary and parameter estimated values are shown in Table 3. Considering the general law that the travel rate decreases with the per capita travel cost and comprehensively summarizing the statistical indicators $R^2$ and F value of each model, the exponential function was selected as the first-stage demand function.

**Table 3.** Regression fitting model of travel rate and per capita travel cost.

| Regression Model | $R^2$ | F | df1 | df2 | Significance Level $p$ | Constant | Coefficient b1 | Coefficient b2 | Coefficient b3 |
|---|---|---|---|---|---|---|---|---|---|
| Linear | 0.175 | 5.286 | 1 | 25 | 0.03 | 0.027 | $-8.779 \times 10^{-6}$ | | |
| Logarithm | 0.163 | 4.874 | 1 | 25 | 0.037 | 0.145 | $-0.018$ | | |
| Reciprocal | 0.135 | 3.900 | 1 | 25 | 0.059 | $-0.007$ | 31.200 | | |
| Quadratic | 0.175 | 2.540 | 2 | 24 | 0.100 | 0.029 | $-1.036 \times 10^{-5}$ | $3.467 \times 10^{-10}$ | |
| Cubic | 0.247 | 2.514 | 3 | 23 | 0.084 | $-0.110$ | 0.0002 | $-9.287 \times 10^{-8}$ | $1.345 \times 10^{-11}$ |
| Power | 0.323 | 11.923 | 1 | 25 | 0.002 | $9.35 \times 10^{10}$ | $-4.113$ | | |
| Exponent | 0.368 | 14.540 | 1 | 25 | 0.001 | 0.184 | $-0.002$ | | |
| Inverse | 0.258 | 8.678 | 1 | 25 | 0.007 | $-9.642$ | 7 044.932 | | |

According to the first-stage demand function and the data presented in Table 2, considering 15 USD as the additional cost, the number of tourists after the additional cost in each source area was determined. Considering the number of tourists as the dependent variable and the additional cost as the independent variable, a curve fitting model was established between them. By comparing the $R^2$ and F values, it was concluded that the inverse function had a higher degree of fit (Figure 3).

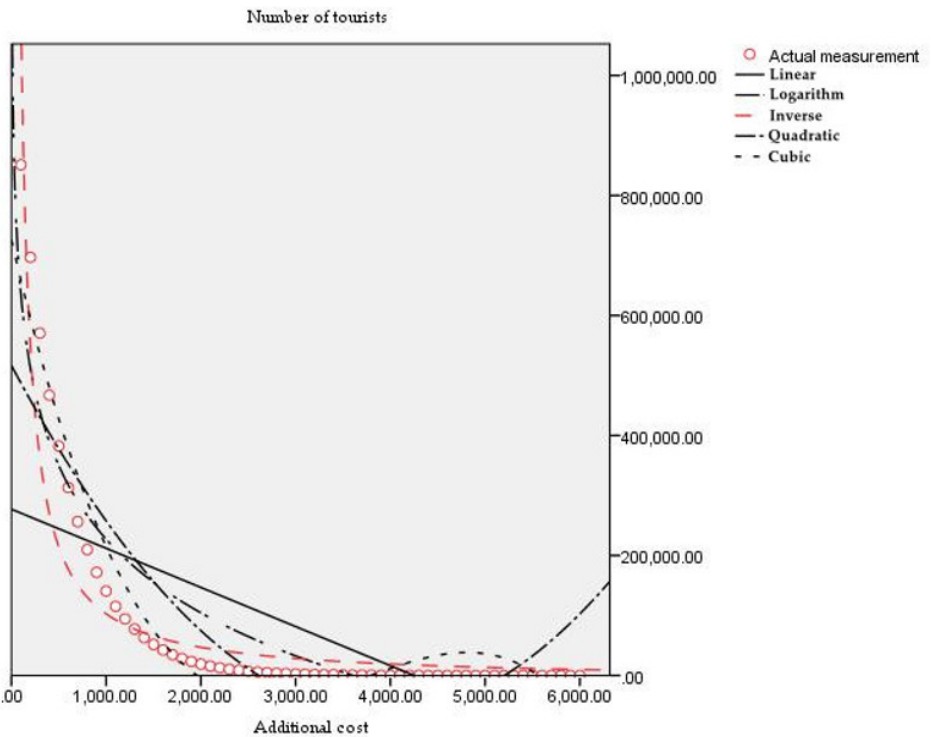

**Figure 3.** Curve fitting for the number of tourists and additional cost.

According to the tourism demand function curve drawn in Figure 3, the travel cost is found to be inversely proportional to the number of tourists. When the total travel cost per capita increases to 900 USD, the corresponding number of tourists will be zero. The area between the tourism demand curve and the X-axis represents the consumer surplus. Integrating the second-stage tourism demand curve function, the consumer surplus of recreational and cultural ecosystem services of Changhai County was 137.43 million USD.

In conclusion, the total recreational and cultural ecosystem services value of Changhai County in 2019 (TV) was 533.14 million USD. The multiple recreational and cultural services values in Changhai County were inferred from the choice behavior and consumption behavior of tourists. According to the types of cultural services proposed by Costanza [61] and referring to the results of the questionnaire, the recreational and cultural services of Changhai County can be divided into five categories: leisure and entertainment, aesthetic experience, educational knowledge, spiritual satisfaction, and social relations. Their multiple values are shown in Figure 4; the aesthetic experience services value was 201.35 million USD, as most tourists chose to travel to Changhai County because they were attracted by the island's ecological environment. The suitable climatic conditions of the island also make it a preferred summer resort; thus, the leisure and entertainment services value was 125 million USD. In addition, the survey results indicated that some tourists only travel to the islands to relax and gain a sense of spiritual satisfaction, accounting for a services value of 118.71 million USD. Moreover, many tourist trips were designed for parents and children, families, couples, and friends, which promoted more harmonious social relationships. The social relationship services were valued at 69.16 million USD. Finally, the islands are rich in

fishing culture and traditional cultural activities; thus, the value of educational knowledge was 18.86 million USD.

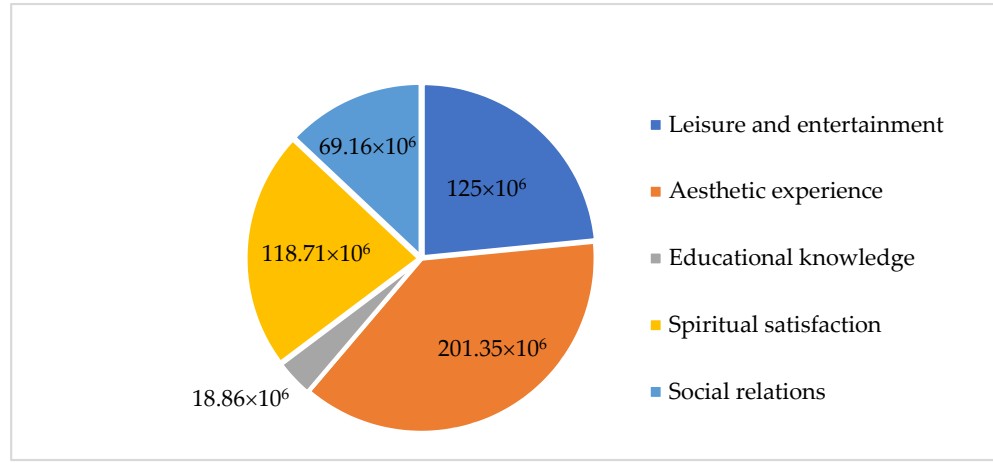

**Figure 4.** The multiple recreational and cultural services values in Changhai County (unit: USD).

*3.3. Assessing the Recreational and Cultural Ecosystem Services Value of the Major Islands in Changhai County*

According to the characteristics of "One Island, One Brand," this research used the ITCM to assess the recreational and cultural ecosystem services value of each key island in Changhai County. Equation (2) was used to calculate the per capita travel cost of each island. The per capita travel costs for Dachangshan, Xiaochangshan, and Guanglu Islands were 285.98, 358.73, and 299.58 USD, respectively. Among them, the Xiaochangshan Island per capita travel cost is relatively high. This is mainly because there are many high star-level hotels on the island, such as SweetHome Apartments, Linyang International Hotel, and Bo'an Jinwan Resort Hotel. The number of trips by individuals traveling to destinations was an essential dependent variable for the ITCM to measure the value of recreational and cultural services. To ensure data accuracy, the model was selected as the average number of trips to each island. The number of trips had significant negative correlations with travel cost and tour time ($p < 0.01$) and a significant positive correlation with the income of the tourist ($p < 0.01$). Tourist satisfaction also affected the number of trips ($p < 0.05$) (Tables 4–6).

**Table 4.** Analysis of the correlation between the number of trips to Dachangshan Island and other factors.

| Variable | Traveling Cost | Tour Time | Gender | Age | Education | Income | Satisfaction |
|---|---|---|---|---|---|---|---|
| Correlation coefficient | −0.337 ** | −0.297 ** | −0.066 | −0.134 | 0.043 | 0.334 ** | 0.206 * |
| Sig. (two-sided) | 0.000 | 0.002 | 0.505 | 0.170 | 0.661 | 0.000 | 0.034 |

** At the level of 0.01 (two-tailed), the correlation is significant; * at the level of 0.05 (two-tailed), the correlation is significant.

**Table 5.** Analysis of the correlation between the number of trips to Xiaochangshan Island and other factors.

| Variable | Traveling Cost | Tour Time | Gender | Age | Education | Income | Satisfaction |
|---|---|---|---|---|---|---|---|
| Correlation coefficient | −0.255 ** | −0.251 ** | −0.116 | 0.201 * | −0.104 | 0.367 ** | 0.194 * |
| Sig. (two-sided) | 0.005 | 0.006 | 0.210 | 0.029 | 0.265 | 0.000 | 0.035 |

** At the level of 0.01 (two-tailed), the correlation is significant; * at the level of 0.05 (two-tailed), the correlation is significant.

Based on the correlation analysis results in Tables 4–6, the fitting conditions were obtained by Equation (6). The probability value of the variable coefficient t test and the

collinearity statistic value [62] were combined to establish the tourism demand function model for each island.

**Table 6.** Analysis of the correlation between the number of trips to Guanglu Island and other factors.

| Variable | Traveling Cost | Tour Time | Gender | Age | Education | Income | Satisfaction |
|---|---|---|---|---|---|---|---|
| Correlation coefficient | −0.295 ** | −0.337 ** | −0.114 | −0.200 | −0.142 | 0.404 ** | 0.256 * |
| Sig. (two-sided) | 0.008 | 0.002 | 0.316 | 0.076 | 0.211 | 0.000 | 0.023 |

** At the level of 0.01 (two-tailed), the correlation is significant; * at the level of 0.05 (two-tailed), the correlation is significant.

The consumer surplus of each island was calculated using Equation (7). The per capita consumer surplus on Dachangshan, Xiaochangshan, and Guanglu Islands were 104.69, 242.23, and 98.23 USD, respectively, indicating that there are considerable consumption opportunities available on Xiaochangshan Island, and tourists have a high willingness to spend. Subsequently, based on the number of tourists received by each island in 2019, the recreational and cultural ecosystem services value of Dachangshan, Xiaochangshan, and Guanglu Islands were calculated; the results are shown in Table 7. Although Xiaochangshan Island had the highest per capita traveling cost and consumer surplus, its recreational and cultural ecosystem services value was the lowest, with only 60.03 million USD, which was far lower than that of Dachangshan Island (294.16 million USD). Because Dachangshan and Xiaochangshan Islands are geographically situated relatively close together, their recreational and cultural ecosystem industries are developing similarly. However, Dachangshan Island, which is the administrative center of the county government, already has a sound infrastructure; therefore, more tourists choose to travel to Dachangshan Island. The recreational and cultural ecosystem services value of Guanglu Island was 143.26 million USD.

**Table 7.** Recreational and cultural ecosystem services value of each island in 2019.

| Island | Per capita Travel Cost (CNY) | Per capita Consumer Surplus (CNY) | Ecosystem Culture Economic Value (CNY) |
|---|---|---|---|
| Dachangshan Island | 285.98 | 104.69 | $294.16 \times 10^6$ |
| Xiaochangshan Island | 358.73 | 242.23 | $60.03 \times 10^6$ |
| Guanglu Island | 299.58 | 98.23 | $143.26 \times 10^6$ |

As is also confirmed in Figure 5, the contribution value of aesthetic experience from each island had the largest proportion. The aesthetic experience value of Dachangshan Island accounted for 34.39%, which was slightly lower than that of the other two islands (the aesthetic experience value of Xiaochangshan and Guanglu Islands accounted for 39.11% and 39.74%, respectively). Between these two islands, the essential tourist attractions for enjoying the beautiful scenery are mainly Xiaoshuikou Forest Park and the Xiaozhu Mountain Site. Dachangshan Island has a larger area, a longer coastline, and a relatively sound infrastructure base; it is an ideal location for tourists to relax. Accordingly, the spiritual satisfaction value on Dachangshan Island accounted for 26.98%, while that of Xiaochangshan and Guanglu Islands accounted 18.55% and 22.44%, respectively. Every summer, many tourists go beachcombing along the Yinniu Bay Golden Coast and at Moon Bay Beach, for amusement. The leisure and entertainment values of Dachangshan, Xiaochangshan, and Guanglu Islands exhibited few differences, accounting for 22.75%, 24.60%, and 22.44%, respectively. The social relationship value of Dachangshan, Xiaochangshan, and Guanglu Islands accounted for a balanced proportion (12.70%, 12.50%, and 14.10%, respectively). Overall, compared with other values, the educational knowledge value of the three islands was the lowest, accounting for 3.17%, 5.24%, and 1.28%, respectively.

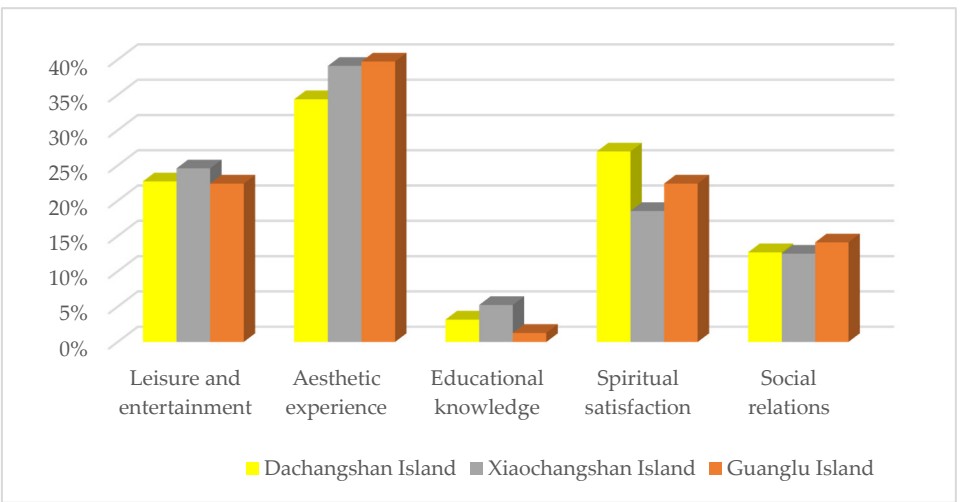

**Figure 5.** Comparison of the proportions of multiple recreational and cultural services values on the three main islands.

## 4. Discussion

### 4.1. Calculation of Consumer Surplus Value

Synthesizing the findings indicated that the ZTCM had two components for calculating consumer surplus. The first component was based on the regression model of travel cost and travel rate to obtain the travel demand function of each tourist source area after additional cost. Then, the consumer surplus in each tourist source area was obtained by integrating these functions. The total consumer surplus in the research area was obtained by adding the consumer surplus in each tourist source area [54,63]. The second component was derived by regression fitting the data on the additional cost and the total number of tourists in each source area to obtain the corresponding function expression, i.e., the actual tourism demand curve equation. Then, by integrating the tourism demand curve equation, the consumer surplus in the destination was calculated, with the lower limit of the integration being zero and the upper limit being the additional cost value when the number of tourists was zero [64–66]. The two methods differ in terms of their conditions. The first applies to the consumers' source area, and the quantity is more important, covering the range of consumers. The second is suitable for when the source of tourists is relatively concentrated, such as the small scope of an island, city parks, and when tourists mainly come from the surrounding area, i.e., for when long-distance tourism is less attractive. Considering the lack of tourist source areas for certain provinces and cities, the final choice was to integrate the overall tourism demand curve function.

### 4.2. Application of the Evaluation Results

Consumer surplus is an economic concept and plays an important role in welfare economics [67]. It refers to the difference between the monetary amount consumers are willing to pay for a commodity and the monetary amount they actually pay for the commodity [68]. According to the calculation results of ZTCM, the consumer surplus in the recreational and cultural services values of Changhai County was 137.43 million USD. For profit, this surplus was still found to be available for development. Consumer surplus is an important reference point for the pricing of tourism products. If the product pricing is greater than the price consumers are willing to pay, the consumer surplus will be negative, and consumers will find it difficult to accept the price, thereby affecting tourism development. If the product's price is lower than the consumer's willingness to pay, tourism sees a profit reduction. Only when the pricing of tourism products is similar to the price consumers are willing to pay, that is, the consumer surplus is zero, is the price acceptable to both consumers and tourism operators, which is conducive to the healthy and sustainable development of tourism [69]. According to the calculation results of the ITCM model, the

per capita consumer surplus on Dachangshan, Xiaochangshan, and Guanglu Islands were 104.69, 242.23, and 98.23 USD, respectively. This standard should be referred to when each island formulates the tourism product price.

The total recreational and cultural ecosystem services value of Changhai County in 2019 (TV) was 533.14 million USD, while the tourism revenue was 241.02 million USD in the same year. According to the comparison, the cultural tourism industry in Changhai County still has considerable scope for development, indicating that the culture tourism industry in Changhai County still exhibits good development [70]. The recreational cultural value of 533.14 million USD of the ecosystem can also be used as the upper limit standard of ecological tourism compensation, and this value can be referred to in the formulation of the standard [71].

According to the results, the concept of ecosystem services should be integrated into subsequent planning, with cultural services as the guide, and by fully considering services, such as aesthetics, recreation, sports, and inspiration. Three recommendations are given, as follows. First, the potential of cultural service resources should be fully realized, the cultural service industry chain should be extended, and culture and participation service programs should be increased. Relying on the good eco-human environment of Changhai County and considering the service experience of fisherman's family tourism as the leading service, a series of special products, such as seascape health preservation, outdoor training, fishing experience, customs activities, sea fishing yachts, and others, could be cultivated. Second, services should be expanded, and the peak tourist season should be extended. The Changhai County islands experience seasonal tourism, with obvious low and peak seasons. The low season mainly enriches cultural services, such as sports, inspiration, scientific research and education, and cultural heritage. Cultural activities should be organized, such as bird seminars, cycling tours, and photography tours. Research and study travel could be used as a key development program in the low season. Third, the tourist source market should be expanded. According to the survey statistics, tourists for recreational and cultural services are mainly from within the province, while tourists outside the province account for a relatively small proportion. Thus, it is necessary to expand the tourist source market beyond the province, carry out long-term cooperation with travel agencies outside the province, and promote Changhai County with the help of the sales channels of these travel agencies.

### 4.3. Research Limitations and Prospects

In the present study, ZTCM was used to analyze the overall recreational and cultural ecosystem services value and the multiple values of Changhai County. ITCM was used to compare and analyze the recreational and cultural ecosystem services values of the county's three main islands. However, this study still has some limitations. (1) In general, the TCM has high requirements on the number and structure of random samples. In this study, the survey period was singular, and the random samples were too concentrated; thus, it was not possible to conduct further research on tourists in the low or peak seasons. We determined that the tourist source areas estimated by the survey were not sufficiently comprehensive, and thus, there was an overall shortage of tourist samples from certain provinces and cities. Therefore, the obtained ZTCM results are somewhat biased and do not accurately reflect the actual situation. The authors will improve upon this limitation in future research. (2) Owing to the intangible and subjective characteristics of recreational and cultural ecosystem services themselves, the economic value analysis results obtained using different methods tend to vary greatly and hence, cannot entirely reflect the social attributes and spatial heterogeneity of such services. A potential target for discussion and research is to combine domestic and foreign research results for a social value assessment of recreational and cultural ecosystem services on islands from the social and spatial perspectives [72–75]. (3) Further, the traditional questionnaire survey method for analyzing recreational and cultural ecosystem services has high time and labor costs. Moreover, with the advent of the era of big data, data sources for assessing the value of recreational

and cultural services have become broader. For example, Richards [76], Van Berkel [77], and Schirpke [78] have all assessed the value of the recreational and cultural services in their research areas based on social media data. In another study, Dai [79] developed a recreational services and sports services model based on cellular signaling data and cellular sports software data and used different analysis methods to assess the cultural ecosystem services value of urban parks and green spaces. Due to the unique circulation closure of islands, an appropriate future research focus would be to incorporate big data into analyses of the various cultural ecosystem services of islands.

## 5. Conclusions

This study innovatively applied both ZTCM and ITCM methods to estimate the value of island ecosystem recreation cultural services in Changhai County, China. First, it estimated the total value and various values of the recreation cultural services in Changhai County from an overall perspective and obtained an overall understanding of the recreation cultural services in Changhai County. Subsequently, three main islands of Changhai County (Dachangshan, Xiaochangshan, and Guanglu Islands) were selected as the study areas, and the recreational and component values of the three islands were compared and analyzed to enable clearer identification of the development status and future development direction of the recreational services of the three islands.

ZTCM was used to analyze the overall recreational and cultural ecosystem services value of Changhai County. The total recreational and cultural ecosystem services value of Changhai County in 2019 (TV) was 533.14 million USD. Among the different services, traveling cost was 395.71 million USD and the consumer surplus was 137.43 million USD. The results show that the values of various recreational and cultural ecosystem service types in Changhai County are ranked according to tourist choice behavior and consumption behavior as follows: aesthetic experience > leisure and entertainment > spiritual satisfaction > social relationships > educational knowledge. These assessment results can provide a scientific basis for elucidating the relationship between cultural development and environmental protection of islands based on a new perspective and promote the coordinated and sustainable development of the economic, social, and ecological benefits of all islands in Changhai County.

ITCM was used to compare and analyze the recreational and cultural ecosystem services values of Dachangshan, Xiaochangshan, and Guanglu Islands. The highest per capita travel cost on Xiaochangshan Island was 358.73 USD. The consumer surplus on Dachangshan, Xiaochangshan, and Guanglu Islands were 104.69, 242.23, and 98.23 USD, respectively. Although Xiaochangshan Island had the highest per capita traveling cost and consumer surplus, its recreational and cultural ecosystem services value was the lowest, with only 60.03 million USD, which was far lower than that of Dachangshan Island (394.16 million USD). Based on the findings of this study, future research should focus on the planning and development of Xiaochangshan Island tourism to maximize consumer surplus.

The multiple values of recreational and cultural services on Dachangshan, Xiaochangshan, and Guanglu Islands should be calculated with reference to tourist preferences. The contribution value of the aesthetic experience from each island had the largest proportion. The aesthetic experience value of Xiaochangshan and Guanglu Islands was as high as 39%. Compared with other islands, Dachangshan Island has the largest proportion of spiritual satisfaction value (26.98%). The leisure and entertainment value of Dachangshan, Xiaochangshan, and Guanglu Islands account for 22.75%, 24.60%, and 22.44%, respectively. Overall, the social relations and educational knowledge cultural services of the three islands had few differences, and these factors occupied only a small proportion of the total value. Thus, based on their own resource characteristics and cultural service advantages, each island could increase the non-material benefits of its ecosystem and develop in harmony with provision, support, and regulation services in their ecosystem.

**Author Contributions:** Conceptualization: N.Z. and H.W.; methodology: N.Z. and J.Z.; software: N.Z. and D.S.; validation: N.Z., H.W., J.Z. and D.S.; formal analysis: N.Z. and H.W.; investigation: N.Z. and H.W.; resources: N.Z. and D.S.; data curation: N.Z. and H.W.; writing—original draft preparation: N.Z., H.W., J.Z. and D.S.; writing—review and editing: N.Z., H.W., J.Z. and D.S.; visualization: N.Z.; supervision: H.W.; project administration: N.Z. All authors have read and agreed to the published version of the manuscript.

**Funding:** This research received no external funding.

**Institutional Review Board Statement:** Not applicable.

**Informed Consent Statement:** Not applicable.

**Data Availability Statement:** Not applicable.

**Acknowledgments:** The authors thank the tourists for their active cooperation in filling out the questionnaire and the government of Changhai County for their support.

**Conflicts of Interest:** The authors declare no conflict of interest. The funders had no role in the design of the study; collection, analysis, or interpretation of data; writing of the manuscript; or in the decision to publish the results.

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
