# Peer review of "Assessment of Recreational and Cultural Ecosystem Services Value of Islands"

_land, doi:10.3390/land11020205_

Round 1
Reviewer 1 Report
The manuscript titled "Assessment of Recreational and Cultural Ecosystem Services Value of Islands" intends to improve the government’s understanding of the island ecological environment system in a more comprehensive manner. Moreover, the study outlines through the combination of improving island facilities and deepening the cultural value of recreational products, the recreational and cultural ecosystem services value of islands can be significantly enhanced.
The research is original; it could be characterized as novel and in my opinion important to the field, it also has almost the appropriate structure and language been used well. In the meanwhile, the manuscript has a good extent (about 6,000 words) and it is comprehensive.
The title is all right. The abstract reflects well the findings of this study, and it has a nice length but avoid use the numbers. The introduction is effective, clear, and well organized; it really introduced and put into perspective what research is negotiating but is too short. Please revise the Introduction of the manuscript and include references which are already exists in bibliography (as you did). Moreover, it does not contain a clear formulation and description of the research problem. Please insert a clear description and justification of the problem the article deals with. The methodology followed is not sufficiently documented and needs to be explained clearly.
The results are very good. The argument flows and is reinforced through the justification of the way elements are interpreted. The same does not applies to the discussion and conclusions sections. Do not forget, Discussion and Conclusion sections should be consistent in terms of Proposal, Problem statement, Results, and of course, future work. Your conclusion section does not justice your work. Make it your key contributions, arguments, and findings clearer. It is recommended to revise Conclusion and of course say something for future work.
Please, revise the lines 19, 20, 112, 220, 236, 249, 251, 258, 261, 263, 266, 267, 269, 278, 315, 321, 322, 328, 374, 435- 449 & of course Table 1,2,7 and make the appropriate currency exchange using dollars ($) or euros (€) or both also you can keep yuan. This is because the results of the research must be directly comparable to other similar surveys that have already been carried out around the world and other such surveys will certainly be carried out, and do not forget, the journal “Land” is international.
Please revise the references of the manuscript and include references which are already exists in bibliography. I would be much more satisfied if the number of references was slightly higher (about 25 - 30 references) and I would appreciate it if also included data from the entire world (Asia, America, Europe and Australia e.tc.). In this way it is documented that a project which is tested in a place with its own characteristics can be implemented in other places around the world.
Please, revise the references, they must have an appropriate style, for this reason I would be good to change [see: Instructions for Authors / Manuscript Preparation / Back Matter / References: - (https://www.mdpi.com/journal/land/instructions or https://www.mdpi.com/authors/references)]. Do not forget, DOI numbers (Digital Object Identifier) are not mandatory but highly encouraged and make the review easier.
Please, revise the reference " 6. Weyland, F.; Laterra, P. Recreation potential assessment at large spatial scales: A method based in the ecosystem services approach and landscape metrics. Ecological Indicators 2014, 39, 34-43. 2014. doi: 10.1016/j.ecolind.2013.11.023.", lines 488-489. I think must be revised as “Weyland, F.; Laterra, P. Recreation potential assessment at large spatial scales: A method based in the ecosystem services approach and landscape metrics. Ecol. Indic. 2014, 39, 34–43, doi:https://doi.org/10.1016/j.ecolind.2013.11.023”.
Please, revise the reference " 13. Tourkoliasa, C.; Skiada, T.; Diakoulakid, D.; et al. Application of the travel cost method for the valuation of the Poseidon tem- ple in Sounio 2015, Journal of Cultural Heritage. 16, 567-574. doi: 10.1016/j.culher.2014.09.011.", lines 502-503. I think must be revised as “Tourkolias, C.; Skiada, T.; Mirasgedis, S.; Diakoulaki, D. Application of the travel cost method for the valuation of the Poseidon temple in Sounio, Greece. J. Cult. Herit. 2015, 16, 567–574, doi:https://doi.org/10.1016/j.culher.2014.09.011.”
Please, revise the reference " 23. Dharmaratne, G; Brathwaite, A. Economic valuation of the coastline for tourism in Barbados 1998, Journal of travel research. 37, 138-144. doi: 10.1016/j.tourman.2013.11.010.", lines 521-522 and use the correct doi: https://doi.org/10.1177/004728759803700205 not the same with reference 22.
Please, revise the references 25 and 37, I think name and surname are confused.
Reviewer 2 Report
Review to Land-1541943
Round 1
Date 6 Jan. 2022
Major comments
The reviewed manuscript title suggests that the study method is a new achievement in general meaning „for islands”, in fact, it is a Xiaochangshan, Dachangshan, and Guanglu islands case study - this regional context must be underlined in the title, abstract and conclusions as well as explained in the main body of the manuscript.
Furthermore, the use of the national currency (CNY) to calculate the ES also emphasizes the regional nature of this research; it is acceptable but at the discussion stage a currency conversion is expected).
The method uses the ZTCM & ITCM approaches, a well-known method of ES valuation, which, taking into account my previous comments, cannot be considered as a novel scientific achievement and contribution to the ES science (rather a kind of well-done homework). This affects my moderate assessment of the peer-reviewed paper originality and novelty. In conclusion - the regional character of the work must be justified, the study cannot generally refer to " islands". A good starting point would be to use L 272-274.
This impression is also maintained by the manuscript introduction section which contains only 23 references and provides rather well known academic knowledge. It looks like intro consumes rather than contribute to the ES science. I do advise Authors to work on this by underlying the novelty of the study and providing a more extended ZTCM & ITCM review, specifically in the context of island ES research.
Minor comments
In the abstract, it is unusual to list „Background”, „Methods” etc.
L31 Human Well-being - shouldn't it be this Subjective Human Well Being (SWB) (e.g. https://doi.org/10.1016/j.ecolecon.2017.12.024)
L66-68 I suppose needs references
L76 – so far I was convinced that the aim of the study was to evaluate the ES; the „government’s understanding’ can be a kind of benefit of the research to the society
L118-113 – can survey method be supported by any references, please?
L131-132 belongs to the results section
L 117-198 (Method section – method should explain how the survey data were managed (any database? geo-survey? mobile application?).
L222-224 belongs to the method sections, L228-229; 274-277; 283-287; 303-310 too. Please improve the manuscript structure, consider using a method flowchart where each step, indices and correlation calculations will be explained to the readers.
Section 4.1 has nothing in common with the scientific discussion – it is a kind of summary and method repetition (not acceptable). The same referees to 4.2 – without references to other authors findings it can not be regarded as discussion. Finally, conclusions should be limited to several most important insights of the study rather than re-calling the ES calculation results.
Reviewer 3 Report
This research made an assessment of recreational and cultural ecosystem services value of Islands. A quantitative process is employed as a methodological tactic. The paper provides an interesting finding. Yet, I still have several comments, which can help the authors improve the manuscript further.
The paper needs a thorough review of the literature. It is quite unusual to present methods directly after the introduction section. This research should include the sound literature review section.
The entire paper is somewhat descriptive. This research needs more in-depth discussion about the findings.
The authors failed to address why this research was necessary. The problem statement is very weakly written.
I wonder what the methodological and theoretical implications of this research are.
Most parts need a professional editing. I found many clumsy sentences.
Nonetheless, the paper deals with an important topic. Thus, my decision is major revision. I look forward to seeing the improved version of the paper.
Round 2
Reviewer 1 Report
The manuscript titled "Assessment of Recreational and Cultural Ecosystem Services Value of Islands" intends to improve the government’s understanding of the island ecological environment system in a more comprehensive manner. Moreover, the study outlines through the combination of improving island facilities and deepening the cultural value of recreational products, the recreational and cultural ecosystem services value of islands can be significantly enhanced.
The manuscript has been revised according the first review comments. The authors carefully studied the comments and revised the manuscript by considering all the comments. All the comments are responded in the new manuscript. I believe the revised manuscript has been improved carefully and I hope the desired level of Land can be reached.
The introduction was modified, conclusions and discussion are better than the previous one, they have general logic and on justification of interpretations as the author’s attribute.
In general, the manuscript is completely different from the previous one, since all the comments of the previous review have been revised.
Reviewer 2 Report
First of all, I would like to thank the Authors for their detailed answers to my comments – all have been appropriately addressed. Secondly, many improvements have been made to the manuscript, and the revised version is far better than the previous one; finally, I recommend this manuscript for further publication.
Reviewer 3 Report
A significant improvement is made. The revised version of the paper looks better than the earlier one. The manuscript only needs editorial error correction and text editing.